# Non-Thermal Plasma Induces Antileukemic Effect Through mTOR Ubiquitination

**DOI:** 10.3390/cells9030595

**Published:** 2020-03-02

**Authors:** Sun-Yong Kim, Hyo Jeong Kim, Haeng Jun Kim, Chul-Ho Kim

**Affiliations:** 1Department of Otolaryngology, Ajou University School of Medicine, Suwon 16499, Korea; ystme@ajou.ac.kr (S.-Y.K.); tomato77kim@naver.com (H.J.K.); bluewing226@gmail.com (H.J.K.); 2Oncoprotein Modification and Regulation Research Center, Ajou University, Suwon 16499, Korea; 3Department of Molecular Science and Technology, Ajou University, Suwon 16499, Korea

**Keywords:** Non-thermal plasma, mTOR, RNF126, ubiquitination, lysosome, leukemia

## Abstract

Non-thermal plasma (NTP) has been studied as a novel therapeutic tool for cancer that does not damage healthy cells. In this study, we show that NTP-treated solutions (NTS) can induce death in various leukemia cells through mechanistic target of rapamycin (mTOR) ubiquitination. Previously, we manufactured and demonstrated the efficacy of NTS in solid cancers. NTS did not exhibit any deleterious side effects, such as acute death or weight loss in nude mice. In the present study, NTS induced cell death in myeloid leukemia cells, including acute myeloid leukemia (AML) and chronic myeloid leukemia (CML). We found that mTOR was downregulated in NTS-treated cells via the ubiquitin-proteasome system (UPS). We also identified ‘really interesting new gene’ finger protein 126 (RNF126) as a novel binding protein for mTOR through protein arrays and determined the role of E3 ligase in NTS-induced mTOR ubiquitination. NTS-derived reactive oxygen species (ROS) affected RNF126 expression and lysosomal dysfunction. These findings suggest that NTS has potential antileukemic effects through RNF126-mediated mTOR ubiquitination with no deleterious side effects. Thus, NTS may represent a new therapeutic method for chemotherapy-resistant leukemia.

## 1. Introduction

Leukemia is a cancer of early blood-forming cells. Overall, leukemia is estimated to account for about 3.5% of all cancers and 4% of cancer-related mortalities [1]. There are four main types of leukemia: acute myeloid leukemia (AML), chronic myeloid leukemia (CML), acute lymphoblastic leukemia (ALL), and chronic lymphocytic leukemia (CLL). Aging is a major risk factor for most human cancers [2]. Studies based on age-period-cohort models have concluded that leukemia incidence increases dramatically with age from 80 to 85 years [3,4]. However, except for ALL, leukemia is the most common childhood cancer [5]. AML, CML, and CLL are all age-dependent, with a median age of around 65 to 72 years at diagnosis [6,7,8]. The success of treatment depends on the type of leukemia and the age of the patient. Therefore, the development of new therapeutic tools against leukemia is necessary.

Mechanistic target of rapamycin (mTOR) is a dual-specificity protein kinase that phosphorylates serine, threonine, and tyrosine residues [9]. As a core component of complexes including MTORC1 and MTORC2, mTOR has critical roles in diverse biological processes, such as cell proliferation, survival, autophagy, metabolism, and immunity [10,11]. Upregulation of mTOR signaling can promote tumor growth and progression through diverse mechanisms including the promotion of growth factor signaling, angiogenesis, glycolytic pathways, lipid metabolism, cell migration, and suppression of autophagy [9,10]. Increased MTORC1 and MTORC2 activities have been reported to play a critical role in leukemia initiation, propagation, and relapse [12,13,14]. In particular, the oncogenic activation of mTOR pathways seen in leukemia patients contributes to chemotherapy resistance, disease progression, and poor prognosis [15,16,17,18].

mTOR inhibitors can be classified into three generations. Although the first generation, allosteric inhibitors (including rapamycin and the chemically similar “rapalogs”), have been tested in many clinical trials, they achieved only modest efficacy when applied as monotherapies in most cancers, including the different leukemias [19,20,21]. The second generation, mTOR kinase inhibitors (TORKIs), are currently in phase 1/2 clinical trials for treatment of solid tumors [22,23,24] but have not yet been tested in leukemia. TORKIs induce apoptosis in a variety of malignant lymphoid cell lines, AML cells [25], and in clinical samples of certain lymphoid neoplasms, with some cases of ALL being particularly sensitive [26,27]. Rapalogs and TORKIs also have some potential in combination therapies [28,29,30,31,32]. Moving forward, it will be important to identify which chemotherapeutics are best used in combination with mTOR inhibitors. The third generation of mTOR inhibitor, RapaLink-1, can overcome resistance to first- and second-generation mTOR inhibitors [33]. Gene mutations may affect the sensitivity of drugs that target the protein encoded by this gene. More than 30 activating mutations of mTOR have been reported in human cancer [34,35]. Indeed, identical drug-resistant mutations have also been identified in drug-naïve patients, suggesting that tumors with activating mTOR mutations might be intrinsically resistant to second-generation mTOR inhibitors [33]. Unfortunately, knowledge of mTOR inhibitors is still quite limited. Therefore, the challenge ahead is to identify the most suitable inhibitor that exerts considerable efficacy with minimal toxicity.

Non-thermal plasma (NTP) is an ionized gas composed of charged particles, electronically excited atoms, molecules, free radicals, and ultraviolet (UV) photons [36]. It is a convergence technology that has been studied as a next-generation cancer therapy. NTP has demonstrated anticancer effects in several different solid tumors, both *in vitro* and in vivo [36,37,38,39], including head and neck cancer (HNC) as shown in our previous reports [40,41]. Inhibition of HNC progression was equally achieved by direct application of NTP spray or as an NTP-treated solution (NTS) on cultured cells or tissues. There are two manufactured forms of NTP: the aforementioned NTP direct spray and NTS. NTP spray is effective as a cancer treatment. However, it cannot be directly delivered to the tumor due to the presence of subcutis and other surrounding tissues. In contrast, NTS allows easy delivery in vivo, while offering similar or even more potent anti-cancer effects [42]. NTS can inhibit HNC progression through mitochondrial ubiquitin ligase activator of NFKB 1 (MUL1)-dependent protein kinase B (PKB/AKT) or heat shock protein 5 (HSPA5) ubiquitination and degradation [42,43]. The major advantage of using NTS in cancer therapy is its cancer cell-specific activity [42,44].

To minimize the danger that misfolded proteins pose to cells, nature has evolved a variety of protein quality control mechanisms that maintain protein homeostasis. Central to such quality control is the close observation of proteins by chaperones [45] and the action of two protein degradation systems: the ubiquitin–proteasome system (UPS) [46] and autophagy driven lysosomal proteolysis [47]. We investigated the involvement of UPS in controlling mTOR turnover. mTOR inhibitors provide a rational basis for the development of therapeutic approaches aimed at mTOR degradation. Ubiquitination is a finely regulated process that ensures tight control of proteins levels, namely via E3 ligases that selectively recognize their substrates [48]. In particular, K48-linked ubiquitination generally programs cells for protein degradation through UPS [49]. E3 ligases are, therefore, considered attractive targets for the development of specific therapies. 

In the present study, we determined that NTS induced leukemia cell death in vivo through mTOR ubiquitination and degradation and did so without obvious side effects. Furthermore, we identified the ‘really interesting new gene’ (RING) finger protein 126 (RNF126) as the E3 ligase that ubiquitinates mTOR. We found that RNF126 could interact with mTOR and directly promote its K48-linked ubiquitination in response to NTS treatment. Our results suggest that NTS could be a novel therapeutic tool for leukemia therapy.

## 2. Materials and Methods

### 2.1. Reagents and Antibodies

MG132 (S2619), Imatinib (CDS022173), Rapamycin (R8781), Everolimus (SML2282), Bafilomycin A1 (B1793), cycloheximide (CHX) (C7698) and N-acetylcysteine (NAC) (A9165) were purchased from Sigma-Aldrich (St. Louis, MO, USA). Antibodies were obtained from several sources. Anti-AKT (9272), anti-p-AKT (Ser473, 9271), anti-B-cell lymphoma 2 (BCL2) (15071), anti-BCL-extra large (XL) (2764), anti-caspase 3 (CASP3) (9662), anti-cleaved CASP3 (9664), anti-glyceraldehyde-3-phosphate dehydrogenase (GAPDH) (5174), anti-HA-tag (3724 and 2367), anti-His-tag (12698), anti-heat shock protein 5 (HSPA5) (3177), anti-lysosomal-associated membrane protein 1 (LAMP1) (9091), anti-microtubule-associated protein 1 light chain 3 beta (MAP1LC3B) (3868), anti-myeloid cell leukemia-1 (MCL1) (94296), anti-mTOR (2983 and 2972), anti-p-mTOR (Ser2448, 5536), anti-Myc-tag (2276), anti-Normal Rabbit IgG (2729), anti-poly(ADP-ribose) polymerase (PARP) (9532), anti-ribosomal protein S6 phosphorylated at the serine 235/236 (p-RPS6) (Ser235/236, 4858), anti-ribosomal protein S6 kinase B1 (RPS6KB1) (2708), anti-p-RPS6KB1 (Thr389, 9234), anti-SQSTM1/p62 (#8025), anti-transcription factor-EB (TFEB) (37785), anti-unc-51 like kinase 1 (ULK1) (6439), anti-p-ULK1 (Ser555, 5869), anti-p-ULK1 (Ser757, 14202), horseradish peroxidase (HRP)-conjugated anti-mouse IgG (7076), and anti-rabbit IgG (7074) were all from Cell Signaling Technology (Beverly, MA, USA). Anti-K48-linked ubiquitin (ab140601), anti-K48-linked ubiquitin (ab140601), anti-cathepsin D (CTSD) (ab6313), anti-cathepsin L (CTSL) (ab133641), anti-MUL1 (ab84067 and ab209263), and anti-RNF126 (ab234812) were from Abcam (Cambridge, MA, USA). Finally, anti-mTOR (SAB2702297) was from Sigma-Aldrich.

### 2.2. Cells

FaDu (American Type Culture Collection, ATCC) and SNU1041 (Korean Cell Line Bank, KCLB), human hypopharyngeal squamous cell carcinoma; SCC15 (ATCC), SCC25 (ATCC), and Cal27 (ATCC), human tongue squamous cell carcinoma; PC12 (ATCC), rat adrenal pheochromocytoma; U251MG (Japanese Collection of Research Bioresources Cell Bank, JCRB) and U87MG (ATCC), human glioblastoma astrocytoma; ASC (ATCC), SNU638 (KCLB), MKN28 (KCLB), and MKN45 (KCLB), human gastric adenocarcinoma; SNU16 (ATCC), human gastric carcinoma; Huh7 (KCLB) and SNU475 (ATCC), human hepatocellular carcinoma; hepG2 (ATCC), human hepatocyte carcinoma; NCI-H1975 (ATCC) and NCI-H1993 (ATCC), human non-small cell lung adenocarcinoma; A549 (ATCC), human lung carcinoma; HCT116 (ATCC), human colon carcinoma; HT29 (ATCC), human colon adenocarcinoma; HT1080 (ATCC), human fibrosarcoma; RD (ATCC), human embryo rhabdomyosarcoma; HeLa (ATCC), human cervix epithelioid carcinoma; MDA-MB-231 (ATCC), human breast adenocarcinoma; U266 (ATCC), human multiple myeloma; HL60 (ATCC), acute promyelocytic leukemia; KG1 (ATCC), acute myeloid leukemia; K562 (ATCC), and human chronic myelogenous leukemia were obtained from the America Type Culture Collection (ATCC, Rockville, MD), the Korean Cell Line Bank (KCLB, Seoul, Korea), and Japanese Collection of Research Bioresources (JCRB, Osaka, Japan). Human head and neck squamous cell carcinomas (MSKQLL1, SCCQLL1, and SNU1483) and human esophageal adenocarcinoma OE33 cells were kindly provided by Dr Se-Heon Kim (Yonsei University College of Medicine, Korea). Human thyroid anaplastic carcinomas (8505C and C643) were kindly provided by Dr Yoon Woo Koh (Yonsei University, Korea).

### 2.3. Plasmids

Wild-type (WT) ubiquitin (Ubi)-HA, K48 Ubi-HA, and K48R Ubi-HA were used as described previously [43]. The promoter for cytomegalovirus (pCMV) mTOR-His was purchased from Sino Biological (Beijing, China, HG11508-CH). We generated four kinds of RNF constructs. The four kinds of RNF constructs (pCMV RNF122-Myc, pCMV RNF126-Myc, pCMV RNF151-Myc, and pCMV RNF175-Myc) were codon-optimized for expression in human, synthesized using a commercial gene synthesis service (BIONICS, Seoul, South Korea), and cloned into the pCMV-Myc vector (Clontech Laboratories, Mountain View, CA, USA, 635689). The RING-domain mutated (C229/232A) RNF126 construct was generated using the Quick-Change site-directed mutagenesis kit (Agilent Technologies) with specific primers using pCMV RNF126-Myc as a template. The sequences of the DNA constructs generated by PCR were systematically verified by DNA sequencing. 

### 2.4. NTS Manufacture 

NTS was manufactured as described previously [43]. The plasma source is equipped with a pair of electrodes made of Al_2_O_3_ (high voltage and ground electrodes, 10 x 40 mm dimension, 2 mm gap between electrodes) isolated from direct contact with plasma by a ceramic barrier. The specifications of the power supply for this system are 2-kV minimum, 13-kV maximum (for NTS preparation, 4-kV was loaded for safety), and a mean frequency of 20 ~ 30 kHz was used. These specifications can vary with the type and amount of gas. In this study, helium (He) and oxygen (O_2_) were used as carrier gases because we previously found that the addition of O_2_ to He plasma improved the efficiency of cancer cell death. NTS was prepared followed by 15 minutes treatment of NTP at 15 mL culture media in absence of serum condition. NTS was freshly prepared in the same manner prior to each experiment. For mouse experiments, NTS was prepared using saline.

### 2.5. RNA Interference Analysis

K562 cells were transfected with 500 pmols of scrambled RNAs or siRNAs using Neon Transfection System (Invitrogen). Scrambled RNAs were used as negative controls (BIONEER, Daejeon, Korea, SN-1001-CFG). Duplexes of siRNA targeting *RNF126* were synthesized by BIONEER. The siRNA sequences are as follows: human *RNF126*, 5′-UGUCUAACCUCACCCUCUA-3′ and 5′-UAGAGGGUGAGGUUAGACA-3′.

### 2.6. RT-PCR

*mTOR* and *RNA18S* gene expressions were estimated using RT-PCR (BIO-RAD, Hercules, CA, USA, T100TM Thermal Cycler). The total RNA in K562 cells was isolated by TRIzol^®^ Reagent (Thermo Fisher scientific, 15596018). cDNAs were synthesized with 5 μg of total RNAs and ReverTra Ace^®^ qPCR RT Master Mix (TOYOBO, Osaka, Japan, FSQ-201) according to the manufacturer’s instructions. PCR primer sequences were as follows: human *mTOR*, 5′-CTGGGACTCAAATGTGTGCAGTTC-3′ and 5′-GAACAATAGGGTGAATGATCCGGG-3′; human *RNA18S*, 5′-CAC GGA CAG GAT TGA CAG AT-3′ and 5′-CGA ATG GGG TTC AAC GGG TT-3′. PCR products were separated by 1% agarose gel, stained with GelRed Nucleic acid gel stain (Biotium, Fremont, CA, USA), and visualized using a ImageQuant™ LAS 4000 (FujiFilm, Tokyo, Japan).

### 2.7. Electroporation

DNA or siRNA electroporation was performed with the Neon Transfection System (Invitrogen, MPK10096). For electroporation with the Neon System, K562 cells were pelleted and resuspended in Resuspension Buffer R (Neon 100 μL kit). 1 × 10^6^ cells were transferred to a sterile 1.5 mL microcentrifuge tube, brought to 100 μL final volume of cell suspension with Buffer R, and mixed with plasmid DNA or siRNA. Cells were pulsed once at a voltage of 1600 and a width of 20. After the pulse, cells were quickly transferred into RPMI 1640 medium.

### 2.8. Protein and DNA Microarray

Protein microarrays were performed using a HuProt human proteome microarray version 4.0 (HuProt^TM^, CDI Laboratories). The protein chip was blocked with 20 mM Tris-Cl, pH 7.5, 150 mM NaCl, 0.1% Tween-20, 5% BSA on an orbital shaker at room temperature for ≥ 2 hours. After rinsing three times with phosphate-buffered saline (PBST) (with 0.1% Tween-20), Cy5-labeled (Expedeon) recombinant full-length mTOR (OriGene, 10 μg of protein diluted in 50 mM Tris-HCl, 10 mM reduced Glutathione, pH 8.0) was added and incubated under a glass coverslip in the dark at room temperature for 1.5 hours. After washing three times with 1x PBST and then three times with Milli-Q water, the microarray was centrifuged for 5 minutes in a 50 mL centrifuge tube. The microarray was scanned with an Axon GenePix 4000B Microarray Scanner (Molecular Devices), and the probe signals were acquired using GenePix Pro6.0 software (Molecular Devices). The probes were considered detectable when the z-scores for both duplicates were over 3.

The DNA microarray experiments were performed using an Affymetrix GeneChip^®^ Gene 2.0 ST Arrays. Briefly, 300 ng of total RNA from each sample was converted to double-strand cDNA. Using a random hexamer incorporating a T7 promoter, amplified RNA (cRNA) was generated from the double-stranded cDNA template though an IVT (in-vitro transcription) reaction and purified with the Affymetrix sample cleanup module. cDNA was regenerated through a random-primed reverse transcription using a dNTP mix containing dUTP. The cDNA was then fragmented by uracil-DNA glycosylase (UDG) and apurinic/apyrimidinic endonuclease 1 (APE 1) restriction endonucleases and end-labeled by terminal transferase reaction incorporating a biotinylated dideoxynucleotide. Fragmented end-labeled cDNA was hybridized to the GeneChip^®^ Gene 2.0 ST arrays for 17 hours at 45 ℃ and 60 rpm as described in the Gene Chip Whole Transcript (WT) Sense Target Labeling Assay Manual (Affymetrix). After hybridization, the chips were stained and washed in a Genechip Fluidics Station 450 (Affymetrix) and scanned by Genechip Array scanner 3000 7G (Affymetrix). The expression intensity data were extracted from the scanned images using Affymetrix Command Console software version 1.1 and stored as CEL files. The whole normalized data were imported into the R programming environment (version 3.0.2), and overall signal distributions of each array were compared by plotting using tools available from the Bioconductor Project (http://www.bioconductor.org) (M2) to check good normalization. After confirming whether the data were properly normalized, differentially expressed genes (DEGs) that showed over 2-fold difference between the average signal values of the control groups and treatment groups were selected manually. In addition, the normalized data of selected DEGs were also imported into R for statistical analysis by t-test and genes with p-value less than 0.05 were extracted as significant DEGs for further study (M2). In order to classify the co-expression gene groups that have similar expression patterns, hierarchical clustering analysis was performed with the MEV (Multi Experiment Viewer) software version 4.4 (http://www.tm4.org) (M3).

### 2.9. Cell Viability Assay

Cells were treated with NTS for 24 hours. This was followed by addition of the Cyto X^TM^ Cell viability assay kit, according to the manufacturer’s instructions (LPS SOLUTION, Daejeon, South Korea, CYT3000). Colorimetric detection of metabolic activity was evaluated by measuring the absorbance at 450 nm, using an Epoch microplate spectrophotometer (BioTek Instruments, Winooski, VT, USA). All conditions were tested in six replicates.

### 2.10. Apoptosis Assay

Quantitative apoptotic cell death by NTS was detected using the Annexin V-propidium iodide (PI) apoptosis detection kit I (BD Biosciences) according to the manufacturer’s instructions. Cells were treated with NTS for 24 hours. The cells were then harvested, washed with cold PBS, and stained with Annexin V-fluorescein isothiocyanate and PI at room temperature for 15 minutes in the dark. Apoptosis was detected using a BD FACSAria III cell sorter (BD Biosciences).

### 2.11. Lysosomal Activity Analysis

Lysosomal activity was determined with LysoSensor^TM^Green DND-189 (Thermo Fisher Scientific, Waltham, MA, L-7535). Briefly, for LysoSensor assay, cells were treated with NTS, cultured with 1 μM LysoSensor probe for 15 minutes, and harvested. Lysosomal activity was detected using a BD FACSAria III cell sorter (BD Biosciences).

### 2.12. Intracellular ROS production Analysis

K562 cells were treated with NTS. After NTS treatment, cells were treated with 10 μM hydroethidine (Thermo Fisher, D-1168) for 30 minutes at 37 °C. Fluorescence-stained cells (1 × 10^4^) were quantified using a BD FACSAria III cell sorter (BD Biosciences, San Diego, CA, USA).

### 2.13. Immunofluorescence

K562 cells were treated with NTS or vehicle control for 8 hours. After incubation, cells were deposited on poly-L-lysine (0.01% v/v) coated coverslips and allowed to adhere for 30 minutes. They were then fixed in 4% paraformaldehyde in PBS for 15 minutes. After four washes in PBS, cells were permeabilized with 0.1% Triton X-100 in PBS for 10 minutes and blocked with 1% BSA in PBS for 2 hours. Cells were incubated overnight at 4°C with primary antibodies against anti-mTOR and anti-RNF126, anti-K48-linked ubiquitin, or anti-MAP1LC3B antibodies. After washing, the coverslips were labeled with the secondary antibodies goat anti-mouse IgG (H+L) Alexa Fluor-488 (Invitrogen, A-11029), goat anti-rabbit IgG (H+L) Alexa Fluor-488 (Invitrogen, A-11034), goat anti-mouse IgG (H+L) Alexa Fluor-546 (Invitrogen, A-11030), and goat anti-rabbit IgG (H + L) Alexa Fluor-546 (Invitrogen, A-11035) for 2 hours at room temperature. The slides were washed and mounted with 4′,6-diamidino-2-phenylindole (DAPI) (Invitrogen, P36931). Images were obtained using a confocal laser microscope (Nikon A1R, Nikon, Tokyo, Japan).

### 2.14. Immunoprecipitation

For preparation of K562 cell lysates for Western blotting, cells were collected 24 hours after transfection and lysed on ice in CHAPS lysis buffer (10 mM Tris-HCl pH 7.4, 150 mM NaCl, 1% CHAPS, 1% Triton X-100, 1 mM EGTA, 1 mM EDTA, 1 mM DTT, and 10% glycerol) supplemented with phosphatase and protease inhibitor cocktail (Thermo Fisher Scientific). Lysed cell pellets were removed by centrifugation for 20 minutes. For immunoprecipitation (IP) of targeted protein, cell lysates were incubated with primary antibody overnight. Lysates were then collected and the same amounts of protein were incubated with protein G agarose beads (Invitrogen, 15920-010) for 4 hours. Samples were washed at least three times with lysis buffer and resuspended in sodium dodecyl sulfate (SDS) sample buffer. After boiling the sample, the protein interaction was determined by Western blot.

### 2.15. Western Blot Analysis

For total cellular lysates, cells were lysed on ice in RIPA buffer (Sigma Aldrich, R0278) containing phosphatase and protease inhibitor cocktail (Thermo Fisher Scientific) for 1 hour at 4 °C. Lysates were run on 8%~12% SDS-PAGE gels and then electrotransferred onto polyvinylidene difluoride (PVDF) membrane. Each membrane was blocked with 5% skim milk for 1 hour at room temperature and incubated overnight with primary antibody (1:1000) at 4 °C. After washing with 0.1% Tween-20 (Sigma-Aldrich, P1379) in Tris-buffered saline (Sigma-Aldrich, T8912), the membranes were incubated with HRP-conjugated secondary antibodies. Proteins were visualized using Amersham ECL Select Western Blotting Detection Reagents (GE Healthcare, Chicago, IL, USA, RPN2235) and detected with ImageQuant™ LAS 4000 (FujiFilm, Tokyo, Japan). Band densities were quantified with ImageJ.

### 2.16. mTOR Ubiquitination Assay

mTOR ubiquitination assays were determined by Ni-NTA affinity isolation as described previously [43]. Briefly, cells transfected with pCMV mTOR-His together with each of the indicated plasmids were washed with PBS, lysed in 200 μL of denaturing lysis buffer (50 mM Tris-HCl, pH 7.4, 0.5% SDS, 70 mM β-meraptoethanol) by vortexing, and boiled for 20 minutes at 95 °C. The lysates were diluted with 800 μL buffer A (50 mM NaH_2_PO_4_, 300 mM NaCl, 10 mM imidazole, pH 8.0, protease inhibitor cocktail, and 10 μM MG132) and incubated overnight with 50 μL Ni-NTA Agarose Beads (Invitrogen, R901-15) at 4 °C. Beads were washed 5 times with buffer B (50 mM NaH_2_PO_4_, 300 mM NaCl, 20 mM imidazole, pH 8.0), and bound proteins were eluted by boiling in a mixture of 5× SDS-PAGE gel loading buffer and buffer C (50 mM NaH_2_PO_4_, 300 mM NaCl, 250 mM imidazole, pH 8.0) (1:4). Thereafter, ubiquitinated mTOR was identified with anti-His (Cell signaling) and anti-HA (Cell signaling) antibodies on Western blot.

### 2.17. Cell Cycle Analysis

K562 cells transfected with Mock or pCMV RNF126-Myc plasmid for 48 hours were harvested, washed with PBS, fixed with ice-cold 70% ethanol overnight at –20 °C. The cells were stained with FxCycle^TM^ PI/RNase Staining Solution (Thermo Fisher Scientific) for 1 hour at room temperature, protected from light, and cell cycle profile was determined by Invitrogen^TM^ Attune^TM^ NxT Flow Cytometer (Thermo Fisher Scientific).

### 2.18. Animal Model

BALB/c nu/nu mice were purchased from Orient Bio Co. Ltd. (SungNam, Korea). Six to eight weeks old female mice were used in this study. Mouse-related experimental procedures and mouse handling were conducted in accordance with the Committee for Ethics in Animal Experiments of the Ajou University School of Medicine (AUMC, IACUC No. 2015-0030). Before use, mice were allowed to acclimatize for at least three days. Mice (n = 6) received 200 μL of NTP-treated saline by intravenous (IV) injection once daily for 2 weeks. Control mice (n = 6) received 200 μL of saline by the same route.

For analysis of blood cell damage by NTS, mouse peripheral blood mononuclear cells (PBMCs) were isolated from the heart using lymphoprep and SepMate (StemCell Technologies) following the manufacturer’s protocol. The SepMate tubes were centrifugated at 1200× *g* for 10 minutes. The mouse PBMCs were then resuspended in DMEM medium containing 10% FBS. Blood cells cytotoxicity was measured by PI staining. For PI-positive control, 100 μM H_2_O_2_ was treated at 37 °C for 1 hour.

### 2.19. Mouse Bone Marrow-Derived Macrophage (BMDM) Isolation

For isolation of BMDM from C57BL/6 mice, femur and tibia were separated. Bone marrow was flushed with serum-free DMEM high glucose medium using 26 1/2 gauge needle. After filtering with a 70 μm cell strainer, cells were centrifuged at 1300 rpm for 3 minutes. Red blood cells were lysed using ACK lysis buffer (Lonza). Bone marrow cells were differentiated for 7 days in DMEM supplemented with 10% FBS, 1% antibiotics, and 20 ng/mL granulocyte-macrophage colony-stimulating factor (GM-CSF) (JW CreaGene). 

### 2.20. Statistical Analyses

Data are expressed as means ± standard deviations (SD). Data were analyzed by ANOVA, followed by Students t test and the Tukey–Kramer multiple comparisons method. Values of *P* < 0.05 were considered statistically significant. All experiments were repeated at least three times.

## 3. Results

### 3.1. Lysosome Inhibition Is Associated with NTS-Induced Leukemia Cell Death

We have previously studied the anticancer effects of NTP or NTS on head and neck cancers (HNC), both *in vitro* and in vivo [42,43,50]. The greatest advantage of NTS in cancer therapy is its cancer cell-specific activity [51,52]. Our earlier work suggested that regulation of lysosomal activity was one of the mechanisms involved in the antitumor effects of NTS in solid cancers [43]. Based on this, we investigated whether NTS could induce antileukemic effects and whether lysosome activity is also involved in NTS-induced death in leukemia cells. NTS effectively killed leukemia cells (Figure 1A and Appendix A). The viability of myeloid leukemia (ML) cells, such as AML and CML, was also affected by NTS. Imatinib was the first signal transduction inhibitor (STI) used in a clinical setting and it functions by preventing a breakpoint cluster region (BCR)-abelson murine leukemia viral oncogene homolog fusion (BCR-ABL fusion) protein from exerting its role in the oncogenic pathway in CML [53,54,55]. However, NTS showed even more potent antileukemic effects than Imatinib (Figure 1B). We have previously shown that NTS induces the accumulation of MAP1LC3B puncta in HNC cells through lysosomal inhibition [43]. Herein, we show that this could be replicated in leukemia cells (Figure 1C). Autophagy- or survival-related proteins were also downregulated following NTS treatment (Figure 1D) and, interestingly, the levels of lysosome activity-related proteins, such as TFEB, LAMP1, CTSD, and CTSL were also decreased in cells treated with NTS for 24 hours (Figure 1E and Appendix A). SQSTM1/p62 can bind to cargo molecules, such as misfolded proteins and their aggregates. Following binding, they are encapsulated together into the autophagosome and ultimately degraded by the lysosome in a suicide-like manner [56,57]. However, SQSTM1/p62 levels were increased and lysosomal activity was inhibited in leukemia cells following NTS exposure (Figure 1F). NTS-induced inhibition of lysosome activity inhibition preceded cell death (Appendix A). Taken together, these results suggest that NTS can exert anticancer effects that are associated with NTS-mediated regulation of lysosomal activity, and that these NTS-mediated anticancer effects can be observed not only in solid tumors but also in blood cancer cells.

### 3.2. NTS Induces mTOR Downregulation in Leukemia Cells

Previously, we examined the anticancer effects of NTS in mice with solid cancers using intra-tumoral injection method [42,43]. However, this delivery mode is not applicable for leukemia. First, the safety of NTS was assessed to determine whether it could induce acute side effects in vivo. Therefore, we administered NTS to nude mice by intravenous injection once daily for two weeks. Intravenous NTS was well tolerated and did not cause acute death or weight loss (Figure 2A,B), consistent with a previous study that also found that NTS had no significant effects in mice or on embryo development in zebrafish, [42]. However, we found that NTS induced apoptosis in AML and CML leukemia cells (Figure 1A, Appendix A) but had no deleterious effect on blood cells (Figure 2C). To further confirm this lack of cytotoxicity, we treated mouse bone marrow-derived macrophages (BMDM) with NTS and found that it did not induce apoptosis or cell death (Figure 2D,E). NTS did not elicit any side effects in normal cells under our experimental conditions. NTS treatment induced a more potent antileukemic effect compared to Imatinib, which is currently in use as a leukemia therapy (Figure 1B).

When a protein has multiple modification sites or is ubiquitinated, these ubiquitinated proteins form a smear and appear at a higher molecular weight area upon SDS-PAGE. Smeared phosphorylated mTOR (p-mTOR) and mTOR bands were detected in high molecular weight areas at 24 hours post NTS treatment (Figure 2F,G). NTS-induced downregulation of mTOR or apoptosis was inhibited by NTS dilution (Figure 2H). NTS induced death in leukemia cells at both low and high cell densities (Figure 2i). Despite NTS having antileukemic effects under high cell density conditions, mTOR bands were still observed as higher molecular weight forms. These data indicate that NTS possesses antileukemic effects and is a novel therapeutic material with no observable side effects in vivo.

### 3.3. RNF126 Act as a Novel Negative Regulator of mTOR

Antileukemic effects and lysosome inactivation are involved in NTS-mediated leukemia cell death (Figure 1, Appendix A). In NTS-treated leukemia cell lines, mTOR bands were observed at a high molecular weight, likely due to post-translational modification (PTM) events, such as ubiquitination, while total mTOR gene expression was not affected (Appendix A). This stability control mechanism of mTOR protein was discovered in 2008. In breast cancer cells, mTOR is targeted for ubiquitination and consequent degradation by binding to tumor-suppressing E3 ligase, FBXW7 [58]. In colorectal cancer, FBX8, another tumor-suppressing E3 ligase, exerts a metastasis suppressor function by degrading mTOR [59]. mTOR can be inactivated either by degradation of mTOR-activating proteins or through direct degradation by E3 ligases. Both of these degradation processes are controlled by the UPS [60]. However, few studies have investigated the ubiquitin E3 ligase of mTOR. For these reasons, we sought to identify novel binding partners of mTOR and selected putative E3 ligase proteins that contained really interesting new gene (RING) domains or homologs of E6AP C-terminus HECT domains. We selected RING finger protein 126 (RNF126) and tested whether it could act as an mTOR E3 ligase (Appendix A). RNF126 interacted with mTOR in leukemia cells (Figure 3A, Appendix A). While NTS treatment induced downregulation of mTOR in cells, RNF126 levels were increased (Figure 3B). RNF126 co-localized with mTOR in cells in an RNF126-dependent manner (Figure 3C), and mTOR protein level was altered in an RNF126-dependent manner (Figure 3D,E). Additionally, mTOR downregulation and loss of leukemia cell viability were associated with RNF126-dependent E3 ligase activity (Figure 3F,G; *RNF126 wild type (WT) vs RING mutant (MT)*). 

RNF126 can ubiquitinate and promote the degradation of tumor suppressor p21 [61]. It can also enhance the AKT signaling pathway [62] in breast cancer and HNC. Other reports have shown that RNF126 targets epidermal growth factor receptor (EGFR) [63] and activation-induced cytidine deaminase (AID) for ubiquitination [64]. Conversely, The Cancer Genome Atlas (TCGA) data suggest low mRNA expression, mutation, and deletion of the RNF126 gene in a variety of human tumors, indicating that RNF126 inactivation may occur broadly in human tumors [65,66]. Intriguingly, tumors with reduced expression of RNF126 are correlated with improved disease-free and overall survival [65,66]. In addition, RNF126 can promote homologous recombination by upregulating BRCA1 expression in breast and ovarian cancers. [67]. Besides BRCA1, other E2F1 target genes, such as apoptosis genes, are also regulated by RNF126. We found RNF126 could induce cytotoxicity in leukemia cells in an E3 ligase-dependent manner (Figure 3G) and cell-cycle analysis showed G2/M phase arrest in RNF126 overexpressing leukemia cells (Appendix A). We conducted gene expression profiling in NTS-treated cells by DNA microarray. Four RNF family proteins that showed upregulation by more than two-fold in response to NTS treatment were selected (Appendix A). We also selected other RNF family proteins whose expression was increased by NTS, including RNF122, RNF126, RNF151, and RNF175. However, RNF126 was found to induce the downregulation of mTOR in leukemia cells (Appendix A). This may explain the observation that NTS-induced leukemia cell death was inhibited in RNF126 knock-down cells (Figure 3H,I). Similarly, mTOR overexpression in leukemia cells was associated with resistance to NTS-induced cell death (Appendix A). This relationship between NTS-induced mTOR downregulation and RNF126 upregulation was observed in several leukemia cell lines (Appendix A). We then examined RNF126 expression levels in several cancer cell lines (Appendix A) and found RNF126 expression levels were reduced, including in leukemia cell lines. RNF126 was heavily suppressed in stomach cancer cells. Low expression of RNF126 was associated with poor survival in the TCGA data (Appendix A). Despite numerous conflicting reports concerning the biological function of RNF126 in cancer, our results indicate that RNF126 is associated with NTS-induced antileukemia effects via mTOR downregulation.

### 3.4. RNF126 Induces Ubiquitination and Proteasomal Degradation of mTOR

We observed that mTOR stability was affected by exogenous RNF126 and NTS-induced RNF126 (Figure 3). We then sought to determine whether RNF126-mediated mTOR downregulation was associated with proteasomal degradation because RNF126 was seen to act as a ubiquitin E3 ligase without alterations to mTOR gene expression (Appendix A). mTOR was downregulated in RNF126-overexpressing leukemia cells. However, RNF126-mediated mTOR downregulation was prevented in proteasome-inhibited cells (Figure 4A). Initially, NTS treatment-induced activation of mTOR in NTS-treated cells. However, mTOR stability was reduced in NTS-treated cells compared to RNF126 knock-down cells (Figure 4B). This may be a protective cellular mechanism. NTS-induced mTOR phosphorylation at early time points affected the interaction of mTOR with RNF126, as RNF126 did not bind to mTOR in cells treated with an inhibitor (Figure 4C). Lysine 48 (K48)-linked ubiquitination is a well-known mechanism of proteasomal degradation [49]. RNF126-mediated mTOR downregulation was inhibited by proteasome inhibitor (Figure 4A). Thus, we investigated whether RNF126 could induce mTOR downregulation via K48-linked ubiquitination. NTS treatment-induced colocalization between the endogenous K48-linked ubiquitinated form and mTOR (Figure 4D). RNF126 strongly induced mTOR ubiquitination in both wild type and K48 ubiquitin-overexpressing leukemia cells. RNF126-mediated mTOR ubiquitination was not induced in the K48 ubiquitin mutant (K48R) (Appendix A). K48-linked ubiquitination of mTOR was induced by NTS treatment (Figure 4E, Appendix A). NTS treatment-induced K48-linked ubiquitination of mTOR in an RNF126-dependent manner (Figure 4F). These results indicate that RNF126-mediated mTOR ubiquitination and degradation are associated with NTS-induced antileukemic effects.

### 3.5. ROS Plays a Crucial Role in NTS-Induced mTOR Ubiquitination

NTS contains several reactive oxygen species (ROS)/reactive nitrogen species (RNS) such as nitric oxide, hydrogen, oxygen, and ozone [42]. Our previous study showed that ROS play a role in NTP/NTS-induced cellular events [43]. Accordingly, NTS-induced ROS alteration appears to be important in HNC cellular signaling events including ER stress, autophagy, lysosome activity, and survival [43]. Therefore, we investigated whether ROS could affect leukemia cell death or mTOR ubiquitination. Mitochondrial ROS were increased by NTS treatment and NTS-induced ROS accumulation was inhibited by N-acetylcysteine (NAC) (Figure 5A). NTS-treatment led to apoptosis of several types of leukemia cells. However, NAC prevented leukemia cell death despite NTS treatment (Figure 5B). Therefore, NTS-induced ROS elevation plays an important role in the antileukemic effects of NTS.

NTS-induced mTOR ubiquitination and degradation were shown to have crucial roles in leukemia cell death (Figure 2; Figure 4). Therefore, we investigated whether there were any associations between NTS-induced ROS, RNF126 induction, and mTOR ubiquitination. ROS inhibition prevented RNF126 expression (Appendix A) and mTOR ubiquitination in NTS-treated leukemia cells (Figure 5C). Based on this, we next tested whether NTS-induced mTOR downregulation was associated with ROS increases and found that NTS-induced downregulation of mTOR/RPS6KB1 signaling was prevented in NAC-treated cells (Figure 5D). In addition, NTS-induced lysosomal activity inhibition was restored in NAC-treated cells (Figure 5E). These results suggest that NTS has potent antileukemic effects against several leukemia cell types, with RNF-mediated K48-linked ubiquitination of mTOR playing a crucial role in these antileukemic effects (Figure 6).

## 4. Discussion

The present study examined whether NTS could be used as a novel therapeutic agent for leukemia and whether mTOR ubiquitination and degradation pathways could function as the antileukemic mechanism of NTS. Of particular note, we identified RNF126 as a novel E3 ligase for mTOR in NTS-treated leukemia cells. RNF126 interacted with mTOR and induced K48-linked ubiquitination. RNF126 expression has been shown to be suppressed in several cancer cell lines, and low expression of RNF126 was associated with poor survival of cancer patients in the TCGA database. Our study revealed two novel findings. Firstly, NTS has potential for use as a new therapeutic agent for treating leukemia. Secondly, RNF126 may represent a novel therapeutic target for treating cancer by targeting K48-linked ubiquitination of mTOR (Figure 6).

Leukemia is a malignancy of the blood system. Although research on the pathogenesis of leukemia is slowly unveiling the mechanisms of the disease that are significant in the ongoing clinical treatment of leukemia, resistance and recurrence of leukemia are still challenging problems that remain to be solved. Recently, long-lived reactive oxygen and nitrogen species (ROS and RNS, respectively), generated by non-thermal plasmas in either gaseous or aqueous forms when primary plasma species (ions, electrons, radicals, and dissociated molecules) interact with a liquid phase, have been developed. These radicals react with other molecules in the gas and liquid phases to produce nitric oxide, hydrogen peroxide and nitrite–nitrate anions [68]. As a result, ROS and RNS are detected in plasma-treated cells. Conventional chemotherapies usually have side effects, such as hair loss and decreased blood cell counts, because these anticancer drugs target rapidly dividing, normal cells. NTP, in contrast, can selectively kill ovarian cancer cells but not normal fibroblasts [69]. Other groups have also succeeded in demonstrating selective killing of cancer cells [70]. NTS did not show any sign of embryotoxicity based on a fish embryo toxicity test (FET) using zebrafish [42] or in a mouse model (Figure 2). Therefore, NTS could be used as a novel therapeutic tool for the treatment of leukemia without damaging normal cells including, importantly, blood cells (Figure 2).

RNF126 protein is ubiquitously expressed in the cytoplasm and the nucleus. It has important roles in various intracellular processes that can be dependent or independent of its E3 ligase activity. As an E3 ligase, RNF126 can directly ubiquitinate the frataxin (FXN) precursor available for mitochondrial import, to control levels of mature frataxin [71]. It is noteworthy that reduced FXN expression can lead to Friedreich ataxia (FRDA), a severe genetic neurodegenerative disease [72]. RNF126 protein is also highly expressed in invasive breast cancer tissues where it targets p21 for degradation, thus promoting breast cancer cell proliferation [61]. High expression of RNF126 is an independent predictor of poor prognosis in invasive breast cancer and is considered a potential biomarker for cancer responsiveness to checkpoint kinase 1 (CHK1) inhibitors [73]. One study has indicated that depletion of RNF126 in breast cancer cells can suppress colony formation and tumorigenicity in mice [74]. RNF126 has been identified as an E3 ligase in the Bag6 complex [75], a multiprotein complex implicated in protein quality control. It is involved in the degradation of mislocalized proteins. As a transcription factor, RNF126 directly interacts with E2F1 to positively promote the transcription of BRCA1 [67]. However, the role of RNF126 in leukemia development is still unclear. Low expression of RNF126 was shown to be associated with poor survival in several cancer patients (Appendix A). RNF126 was increased by NTS and was also associated with NTS-induced degradation of mTOR through K48-linked ubiquitination (Figure 3 and Figure 4). Such NTS-induced p-mTOR might interact with RNF126 and lead to ubiquitination and degradation because RNF126 was unable to bind to mTOR in mTOR inhibitor-treated samples (Figure 4C). RNF126 overexpression could lead to cell growth inhibition (Figure 3G) and G2/M cell-cycle arrest (Appendix A). The present study also showed that NTS-mediated lysosomal regulation could be a new strategy in cancer therapy without damaging normal cells.

Among the many functions of mTOR previously listed, mTOR can indirectly inhibit apoptosis [76] through a mechanism that depends on the cellular context and the control of target molecules such as B-cell lymphoma 2 (BCL-2), BCL-2-associated death promoter (BAD), p53, p21, p27, and c-myc [77]. Several research groups have shown that high-level mTOR expression is needed to control apoptosis by BCL-2 family members, thus promoting tumor cell survival [77,78]. The activated PI3K/AKT/mTOR pathway can decrease the BCL-2 homology domain (BH3) mimetic effectiveness in cancer cells by upregulating anti-apoptotic BCL-2 family members, such as myeloid cell leukemia-1 (MCL-1) and B-cell lymphoma-extra large (BCL-XL) [79]. A similar approach has been used by Rahmani et al. [80]. Combined treatment with clinically relevant PI3K and BCL-2 inhibitors may prove effective in the treatment of AML.

p53 can inhibit mTOR by regulating a pathway that is used to detect nutrient deprivation, such as AMPK, and subsequently the TSC1/TSC2 complex [81]. More recently, it has been clarified that products of two p53 target genes, Sestrin1 and Sestrin2, can negatively regulate mTOR signaling by activating AMPK, which in turn phosphorylates TSC2 [82]. Independent of direct interaction between p53 and mTOR in cell death control, the effectiveness of their simultaneous modulation has been highlighted in leukemia. For example, the reactivation of p53 by Nutlin-3 and the inhibition of AKT/mTOR by tanshinone IIA exhibit a synergetic antileukemia effect with Imatinib in Philadelphia-positive ALL [83]. 

Our previous study has shown that NTP/NTS can induce AKT ubiquitination through MUL1 in HNC [42]. Similar results were observed in leukemia cells (Appendix A). If NTS could regulate MUL1 or RNF126 activity at the same time, NTS will be a strong therapeutic tool for leukemia therapy compared to other drugs (e.g., Imatinib) through AKT or mTOR ubiquitination. The present study confirmed that NTS possesses more potent cytotoxicity than Imatinib (Figure 1B). Currently, Imatinib is the standard first-line care in CML. However, various studies have reported that a major drawback of Imatinib is the development of resistance, which is therapeutically challenging [84,85]. Therefore, research has been conducted to develop second and third generations of tyrosine kinase inhibitors to overcome drug resistance without side effects [86,87,88]. For these reasons, NTS could be a new therapeutic stratagem for leukemia therapy with minimal side effects, and low cost to patients. 

In the present study, for the first time, we found that RNF126 acts as a novel E3 ligase for mTOR through K48-linked ubiquitination induced by NTS. NTS could be a new therapeutic for several kinds of leukemia such as CML and AML. ROS play an important role in not only the antileukemic effects of NTS but also in RNF126-induced mTOR ubiquitination (Figure 5). Further experiments should be designed to investigate ubiquitination sites on mTOR by RNF126. Using NTS or targeting RNF126 could be a beneficial therapeutic approach for leukemia therapy.

## 5. Conclusions

We have provided the first demonstration that NTS-induced mTOR degradation is mediated by RNF126. In addition, our study offers strong evidence that RNF126 is a novel tumor-suppressive E3 ligase that regulates mTOR in leukemia cells. Based on these results, we suggest that NTS could be used as a novel therapeutic for leukemia, which acts through mTOR K48-linked ubiquitination and degradation in a phosphorylation-dependent manner.

## Figures and Tables

**Figure 1 cells-09-00595-f001:**
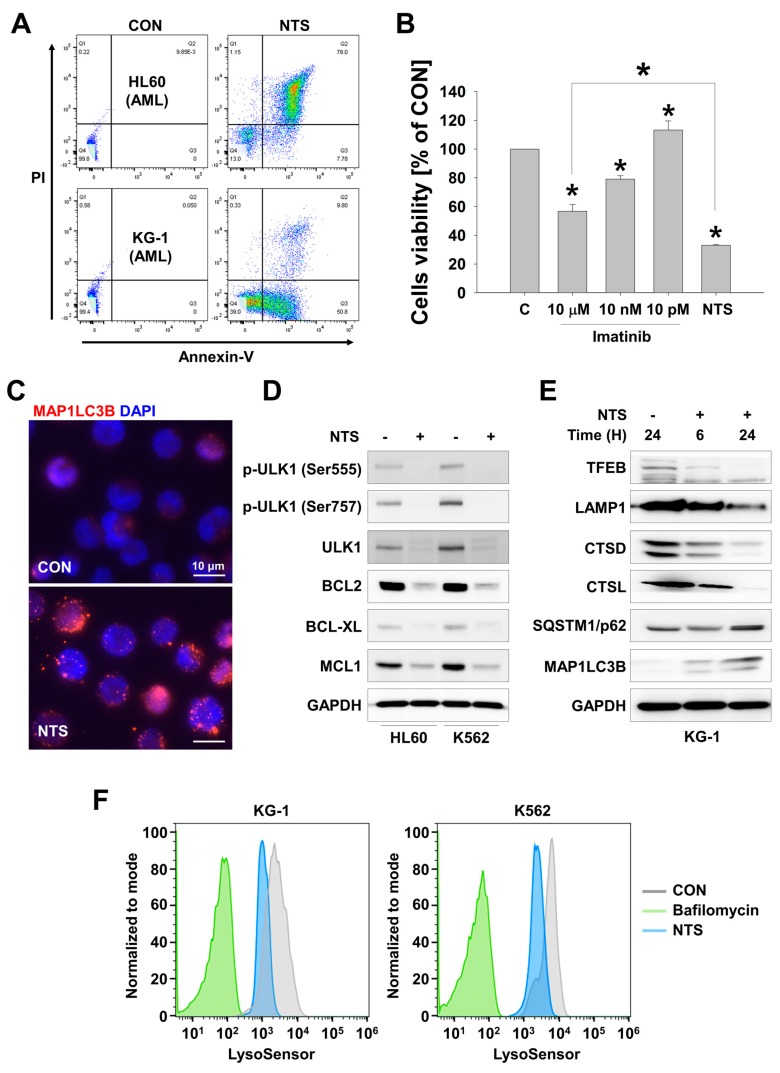
Non-thermal plasma-treated solution (NTS) induces leukemia cell death through lysosomal inhibition. (**A**) FACS analysis of NTS-induced leukemic cell death in HL60 and KG-1 AML cell lines. Cell death was determined by Annexin V/PI staining. (**B**) Antileukemic effects of NTS versus Imatinib in HL60 cells. Cell viability was determined by 3-(4,5-Dimethylthiazol-2-yl)-5-(3-carboxymethoxyphenyl)-2-(4-sulfophenyl)-2H-tetrazolium (MTS) assay. Data are presented as mean ± SD. n = 6, * *P <* 0.05. (**C**) NTS-induced MAP1LC3B accumulation in leukemia cells. NTS-induced autophagy, indicated by MAP1LCB puncta, was visualized using immunocytochemistry. Scale bar-10 μm. (**D**) Western blot analysis of autophagy- or mitochondrial function-related signaling pathways affected by NTS. (**E**) Western blot analysis of lysosomal protein expression in KG-1 cells treated with NTS for 6 or 24 hours. (**F**) FACS analysis of lysosomal activity in KG-1 (AML) and K562 (CML) cells treated with NTS. Bafilomycin (100 nM, 24 hours) was used as a positive control. CON: control.

**Figure 2 cells-09-00595-f002:**
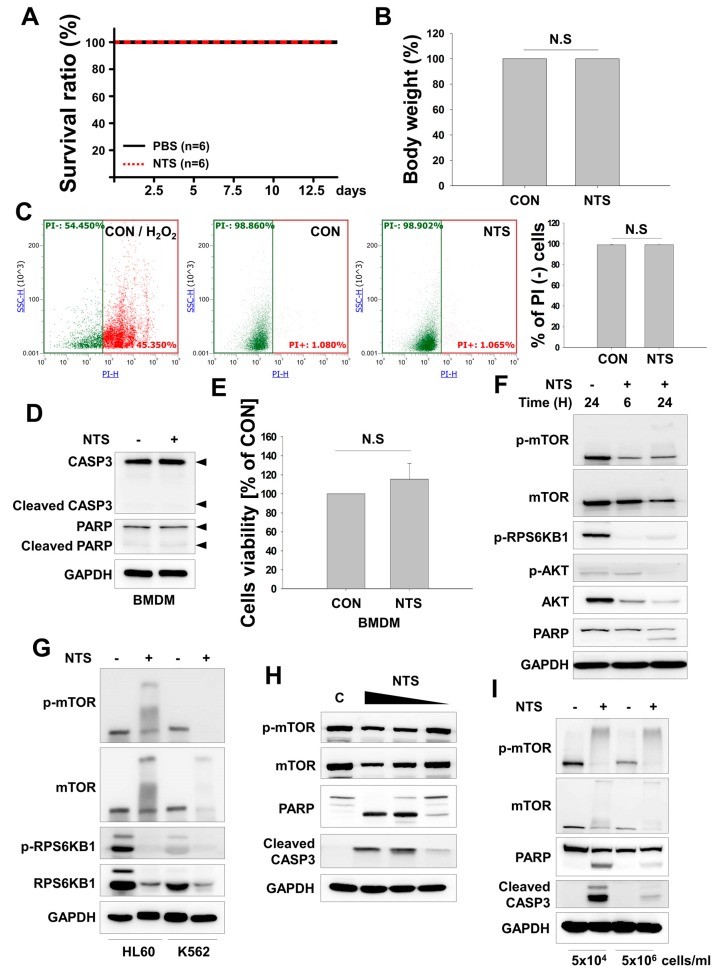
NTS induces downregulation of mechanistic target of rapamycin (mTOR) without acute side effects. (**A**,**B**) NTS was given to mice by IV injection (200 μL) once daily for 2 weeks and acute death (**A**) or loss of body weight (**B**) were monitored. (**C**) FACS analysis of NTS-induced cytotoxicity in peripheral blood mononuclear cells (PBMCs) isolated from control or NTS-treated mice (n = 6). Data are means ± SD. (**D**,**E**) NTS does not induce cell death in mouse bone marrow-derived macrophages (BMDM). (**F**) mTOR expression in KG-1 cells treated with NTS as evaluated by Western blot. (**G**) Western blot analysis of mTOR expression and ubiquitination in NTS-treated HL60 or K562 cells. Ubiquitination is seen as smearing and appearance of the band at a high molecular weight. (**H**) The antileukemic effects of NTS were reduced in a concentration-dependent manner. NTS was serially diluted in serum-free media and used to treat for 24 hours. Protein levels were determined by the indicated antibodies. (**I**) NTS affects the viability of leukemia cells not only in low but also in high cell densities. K562 cells were treated with NTS for 24 hours. Proteins level was evaluated by Western blot.

**Figure 3 cells-09-00595-f003:**
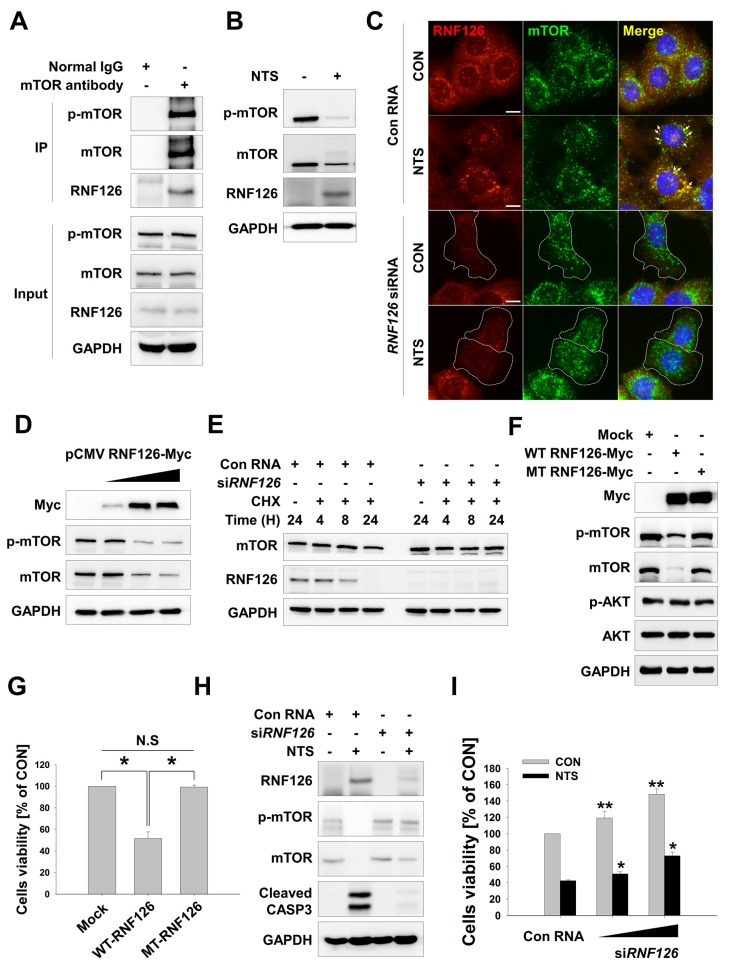
‘Really interesting new gene’ finger protein 126 (RNF126)-induced mTOR downregulation. (**A**) Endogenous interaction between RNF126 and mTOR in leukemia cells as determined by immunoprecipitation assay. Normal IgG antibody was used as a negative control. (**B**) p-mTOR, mTOR or RNF126 protein expression in K562 cells treated with NTS and determined by Western blot. (**C**) Colocalization between RNF126 and mTOR in NTS-treated K562 leukemia cells. RNF126 (red) or mTOR (green) was stained with anti-RNF126 or anti-mTOR antibodies. Scale bar-10 μm. (**D**) Downregulation of mTOR in RNF126-overexpressing K562 cells. RNF126 plasmids were transfected into K562 cells and after 24 hours, protein levels were assessed by Western blot. (**E**) Western blot of enhanced mTOR protein level in siRNF126 knock-down K562 cells. Cycloheximide (CHX, 50 μg/ml) was pre-treated for 1 hour. (**F** and **G**) Western blot of mTOR downregulation and leukemia cell growth inhibition in K562 cells transfected with RNF126 wild type (WT) and E3 ligase inactive mutant (MT) (**F**) MTS assay showing cell viability in K562 cells transfected with RNF126 wild type (WT) and E3 ligase inactive mutant (MT) (**G**, n = 6). (**H** and **I**) RNF126 knock-down affects NTS-induced mTOR downregulation and cell death. RNF126 siRNA was transfected into K562 cells for 48 hours. After then, NTS was treated further 24 hours. Proteins level was evaluated by Western blot using each indicated antibodies (**H**) and cell viability was measured by MTS assay (**I**, n = 6).

**Figure 4 cells-09-00595-f004:**
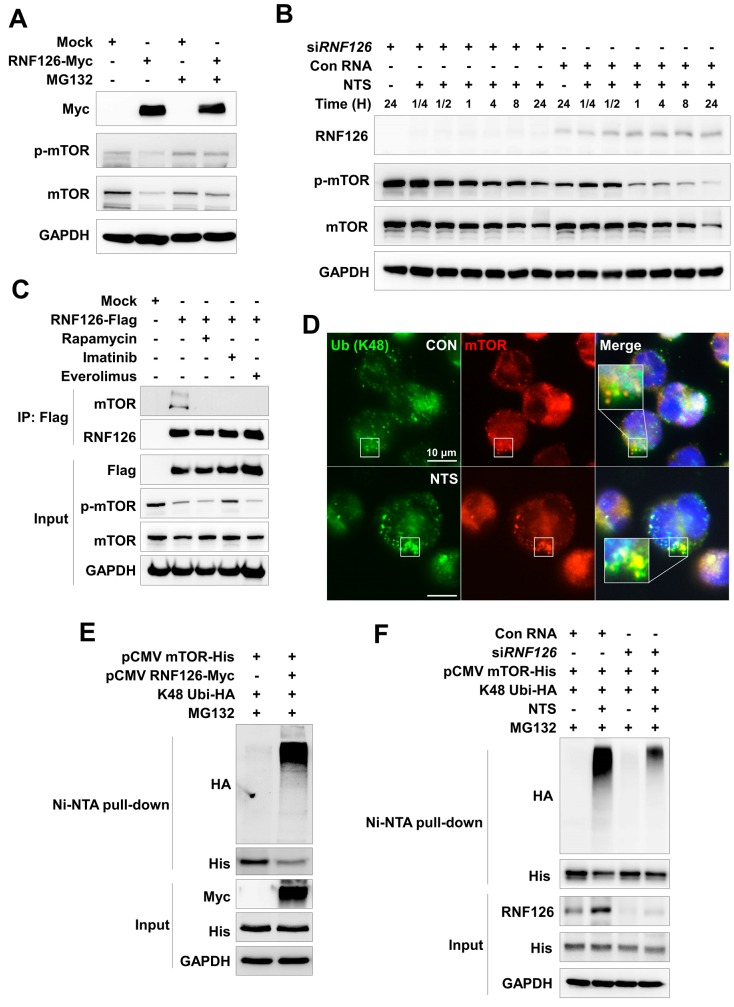
RNF126-induced K48-linked ubiquitination of mTOR. (**A**) Western blot of RNF126-induced proteasomal degradation of mTOR. RNF126 plasmids were transfected into K562 cells before treatment with proteasome inhibitor MG132 (10 μM). (**B**) Western blot of early NTS-induced mTOR phosphorylation in K562 cells transfected with siRNF126 compared to non-transfected control cells. NTS induces degradation of p-mTOR and mTOR through RNF126 dependently. (**C**) Interaction between mTOR and RNF126 evaluated by immunoprecipitation assay. K562 cells were transfected with RNF126 plasmids then treated with mTOR inhibitors, Rapamycin, Imatinib, or RAD001 for 24 hours. (**D**) NTS-induced K48-linked ubiquitination of mTOR in K562 cells treated with NTS. Endogenous K48-linked ubiquitin (green) or mTOR (red) were stained with anti-K48 ubiquitin or anti-mTOR antibodies. Scale bar-10 μm. (**E**) RNF126-induced K48-linked ubiquitination of mTOR in K562 cells transfected with each indicated plasmid and treated with MG132 (10 μM) proteasome inhibitor. mTOR ubiquitination was determined by Ni-NTA His pull-down assay. (**F**) Ni-NTA His pull-down assay of mTOR ubiquitination in RNF126-suppressed cells. The indicated plasmids were transfected and then NTS was added for a further 24 hours. MG132 (10 μM) was added for 6 hours before cell harvest.

**Figure 5 cells-09-00595-f005:**
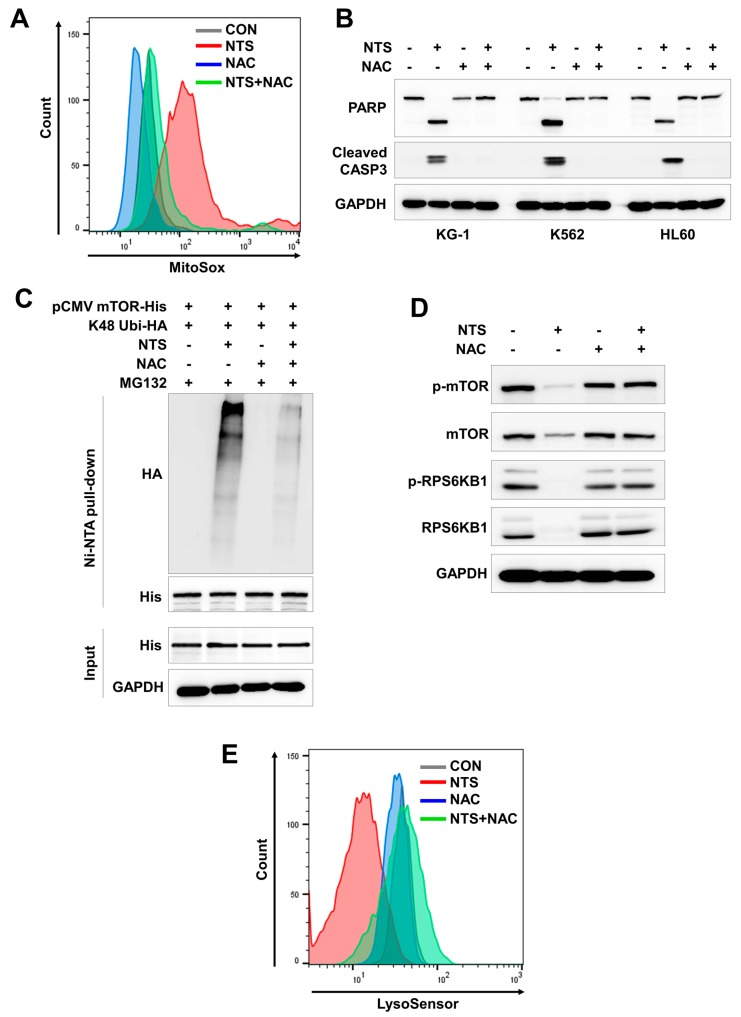
Antileukemic effects of NTS-induced ROS are mediated through mTOR ubiquitination. (**A**) NTS-induced increase in levels of mitochondrial ROS. ROS levels were determined by FACS analysis using the MitoSox in K562 cells treated with NAC prior to NTS treatment. (**B**) Western blot of cleaved PARP and CASP3 in K562 cells treated with NAC prior to NTS treatment. NTS-derived ROS plays a crucial role in leukemia cell death. (**C**) ROS inhibition leads to failure of mTOR ubiquitination. mTOR ubiquitination determined by Ni-NTA His-pull down ubiquitination assay in K562 cells transfected with the indicated plasmids, pretreated with NAC prior to NTS treatment and proteasome inhibition with MG132 (10 μM). (**D**) NAC prevents NTS-induced downregulation of mTOR/RPS6KB1 signaling. (**E**) Lysosome activity is affected by NTS-derived ROS. NAC was pre-incubated for 1 hour and then NTS was added for a further 24 hours. After 24 hours, lysosome activity was assessed by FACS analysis using LysoSensor.

**Figure 6 cells-09-00595-f006:**
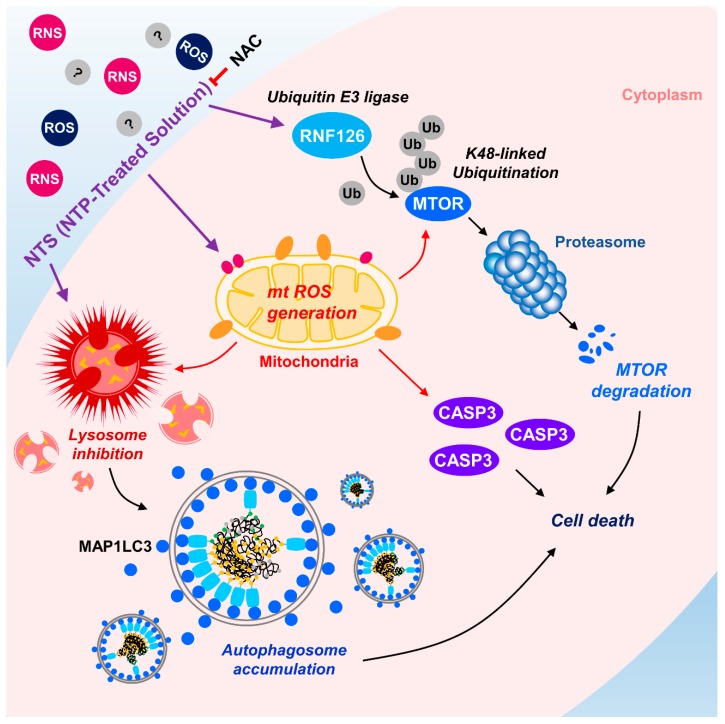
Schematic diagram of RNF126-mediated mTOR ubiquitination in NTS-treated leukemia cells. NTS induces activation of RNF126-mediated mTOR ubiquitination and degradation. NTS-induced mTOR degradation affects lysosomal activity, resulting in accumulation of autophagosomes in cells and subsequent apoptosis. NTS-induced antileukemic effects and/or mTOR ubiquitination are inhibited by antioxidant, NAC.

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
