# Peer review of "Non-Thermal Plasma Induces Antileukemic Effect Through mTOR Ubiquitination"

_cells, 2020, doi:10.3390/cells9030595_

Round 1
Reviewer 1 Report
In the submitted manuscript the authors hypothesize that Non-Thermal Plasma induces antileukemic effect through MTOR ubiquitination.
I cannot comment on biophysical aspects of NTP generation, as I am molecular biologist/biochemist. I will therefore limit myself to comment that part of the manuscript.
Understanding molecular mechanisms of NTP is very important and clearly identifying proteins and processes that mediate the cellular effects is of high relevance.
Lane 280: “NTS induces autophagy in leukemia cells”
I would be careful stating that NTS induces autophagy, as fusion of autophagosomes with lysosomes is inhibited due to lysosome inhibition. It is true that p62 and LC3B accumulate, though. I would therefore rather state: “NTS induces LC3B accumulation in leukemia cells”.
The authors state: “In NTS-treated cells, MTOR bands were observed in high molecular weight areas due to post-transcriptional modification (PTM) events such as ubiquitination.”
That is overstatement: the only thing that can be concluded is that MTOR can be found in high-molecular weight complexes, most probably due to POSTTRANSLATIONAL modifications (not posttranscriptional!!), such as ubiquitination. Please rephrase.
The authors do not provide any info about protein array used for identifying MTOR interacting partners: that needs to be included in materials and methods. Also, they mention DNA microarray generated upon NTS treatment, but raw data is not provided.
The authors state: “We selected RNF126 (ring finger protein 126) and tested whether it could act as a MTOR E3 ligase through interaction.“ That statement is misleading: they cannot test if RNF126 is E3 ligase for MTOR by checking their interaction. Interaction suggests that MTOR might be a substrate of RNF126, and substrate can only be determined by ubiquitination assays. Please rephrase.
The authors need to demonstrate that RNF126 E3 ligase activity (and not only the presence of RNF126) is responsible for MTOR degradation (Figure 3E, 3F, by also including E3 ligase inactive mutant of RNF126), as there is also a possibility that RNF126 acts as scaffold and recruits some other E3 ligase to MTOR.
The authors also need to better resolve the time frame of MTOR degradation and activation/inactivation by RNF126 and NTS, such as 0, 15, 30, 60, 120, 180 min, 6, 12, 18, 24 hours and present it on the same blot.
Is RNF126 E3 ligase activity affected by ROS? The data imply that it is increased, either directly or indirectly upon ROS increase.
Supplementary figure S3: what are the levels of MTOR, LC3B and p62 in these cell lines? Any correlation with RNF126 levels?
English has to be checked by the native speaker, as there are too many mistakes throughout the manuscript such as:
Lane 277: NTS has strong antileukemic effects rather than Imatinib.
Lane 278: NTS or Imatinib was treated for 24 hours.
Lane 279: Cell viability was performed by MTS assay.
Lane 540: NTS could be a new therapeutic stratagem for leukemia therapy without causing side effect together with low cast effect to patients.
Lane 457: NTS-induced MTOR degradation is affected to lysosomal activity, therefore, autophagosome is accumulated in cells and undergoes to apoptosis.
Supplementary figure 1: NTS induces several kinds of leukemia cells (what does that sentence mean??)
Minor comments:
Lane 22: MTOR, not MOTR
At multiple positions “micro” is written wrongly (unusual symbol)
Reviewer 2 Report
Overview:
The authors propose the use of non-thermal plasma (NTP) and NTP-treated solution (NTS) to treat leukaemia and suggest that the mode of action is via ubiquitination of mTOR. NTP/NTS has previously been shown to have an effect on head and neck cancers and this is the first time that it has been shown to have a positive benefit on leukaemia. In addition, the authors use protein and DNA arrays to find novel binding partners of mTOR
Main impression:
While I understand that the authors wish to show the benefit of NTP/NTS on leukaemia, I feel that the manuscript is much broader than the title suggests. I find that there are parts of the manuscript that are very repetitive and information is included that is not relevant to the data generated. I find it surprising that the authors draw conclusions regarding the anti-leukemic effect when only testing a subset of cells, and being inconsistent with which cells are tested in each assay. For example, line figure 2 utilises AML cell lines while figures 5 and 6 test only CML cell lines. It would be interesting to see if the all leukaemia cell lines function in the same way.
Specific points:
1. There are a number of typing and grammatical errors throughout e.g. the μ symbol is frequently given as an @ symbol and mTOR is consistently referred to as MTOR. I suggest that the manuscript is more extensively proof-read, possibly by a native English speaker. Some examples are:
line 386 – “RNF126 did not bound in MTOR inhibitor treated cells” would be better if written as RNF126 did not bind to mTOR in cells treated with inhibitor.
line 392 – “K48R ubiquitination” it is my understanding that this is a typo as K48R is not referred to again in the manuscript
line 422 – “NTS induced increase level of intracellular ROS…” I assume that the authors mean that NTS induced the level of ROS to increase?
2. In section 3.3, the reasons for selecting to focus on RNF126 are explained. However, I feel that this section is laborious and could be better explained. Indeed, it is not until the figure 3 legend (line 362) that the reader is told that the information came from a protein array. In addition, the authors make use of a DNA microarray (line 349), around which some key findings were made which are very interesting. However, these assays are not detailed in the materials and methods, nor are other papers referenced. From the latter experiment, it is stated that the anti-leukemic effect of RNF126 is achieved via the downregulation of mTOR, although I do not think that this can be fully stated as the authors only test 1 cell line for leukaemia and it would be nice to see the same result confirmed in other leukaemia cell lines.
3. The listing of the supplementary figure 2A, B and C in the main text do not match the figures in the supplementary files. It is my opinion that the supplemental figure 2A should be included in the main figure.
4. The legend to figure 5 needs to be clarified (e.g. line 446 “indicated leukaemia cells” but there are no indicated cell lines) and each part of the figure does not match the description in main text.
5. There are also some areas which lack detail, for example in section 3.4 (lines 378-415) the term “early period” is used to refer to both 5, 15 and 30minutes. It would be more informative to tidy this up and relate "early period" to the time frame of interest.
Reviewer 3 Report
Manuscript Non-Thermal Plasma Induces Antileukemic Effect Through MTOR Ubiquitination from Sun-Yong Kim represents an interesting study that presents novel ubiquitin ligase involved in stability of mTOR protein. Such an important discovery has to be supported by clear and undoubtful results. In my view, there are several gaps that have to be filled before publication.
Fig. 1.: NTS induced leukemia death via lysosome inhibition
It is necessary to prove that lysosome inhibition precedes cellular death. It has to be shown that inhibition of cell death (using caspase inhibitors) has no effect on lysosomal inactivation.
It is hard to understand why almost all of the detected proteins are disappearing after NTS treatment (especially when autophagy is inactive). Again, how apoptosis inhibition would affect the disappearance of these proteins? Would it be blocked by proteasome inhibitors?
Experiments lack positive controls. In 1F control like Bafilomycin should be included.
Fig.2.: NTS induces downregulation of mTOR without acute side effects.
It is not clear how much of NTS was given to the mice (dosage).
Analysis of some sensitive tissues like a liver and blood cells should be tested for apoptosis and necrosis.
Again is mTOR downregulation result of apoptosis?
Fig.3.: RNF126 induces MTOR downregulation.
I am missing endogenous interaction. Using either RNF126 or mTOR antibodies.
Is mTOR mRNA level not changing?
The stability of mTOR should be tested using chase assay (e.g. cycloheximide).
The specificity of the RNF126 antibody (for use in IF - 3D) should be tested using siRNA.
Fig.4.: RNF126 induces K48-linked ubiquitination of mTOR
It has to be clarified that the denaturation buffer was used in 4D a 4F. If not, authors have to show that it is without any doubts ubiquitinated forms of mTOR (shown in the upper panel in Fig. 4D a 4F).
Round 2
Reviewer 1 Report
The manuscript has been significantly improved.
Sentence “In NTS-treated cells, mTOR bands were observed in high molecular weight areas due to post-translational modification (PTM) events such as ubiquitination because mTOR gene expression was not changed” should be improved: “In NTS-treated cells, mTOR bands were observed in high molecular weight areas, most probably due to post-translational modification (PTM) events, such as ubiquitination. At the same time, mTOR gene expression was not affected (Supplementary Figure S2A)”
Comment (v2): I do not understand this explanation: “Based on our previous studies, we concluded that ROS did not affect E3 ligase activity but affected the expression level of RNF126. This was because ROS inhibition did not induce mTOR ubiquitination through RNF126 suppression.”
The authors need to clearly show that NTS treatment affects RNF126 gene expression to support their claim. Also, the fact that ROS inhibition did not induce mTOR ubiquitination does not prove that ROS does not affect E3 ligase activity of RNF126 (independent of its effect on RNF126 expression).
What is the proof for this (quite hard to understand) statement: “Oxidative stress or ROS is a kind of cellular stress condition in ubiquitination mechanisms that deal with ubiquitin-related gene expression.”? Any references?
My old comment was: Supplementary figure S3: what are the levels of MTOR, LC3B and p62 in these cell lines? Any correlation with RNF126 levels?
I assume that there was some misunderstanding regarding my question about S3B here. I basically asked the authors to provide blots for expression levels of MTOR, LC3B and p62 in these cell line (for which they show RNF126 levels), to demonstrate if indeed high levels of RNF126 lead to increased ubiquitination and degradation of MTOR, as well as to see how it affects LC3B and p62 levels.
Not all the figures are mentioned in the text, such as S2B – please adjust that.
Scientific English should be edited, as there are few wrong scientific descriptions, such as:
“mTOR ubiquitination assays were determined by Ni-NTA affinity isolation.” Please read the text carefully and correct these items, as probably English translator is not familiar with precise scientific phrasing.
Comment Lane 297 (v2): it is unclear how often in 2 weeks treatment was performed: “Mice (n=6) received 200 μl of 297 NTP-treated saline by intravenous (IV) injection for 2 weeks. Control mice (n=6) received 200 μl of saline by the same route.” Also “proteins levels” and “protein’s stability”. Also “Western blot” and “western blot” – please unify.
Comment: You state: Figure 2D, 2F 2H: NTS does not induce cell death in mouse bone marrow-derived macrophages (BMDM). However, Figure 2F/2H show both PARP and Casp3 cleavage. What is the duration of NTS treatment in Figure 2D? Please clarify.
Reviewer 3 Report
The manuscript is improved. There are several results that have to presented adequately.
1F - Bafilomycin should be added in the same experiment as NTS (that is a minor problem)
3A - other loading control then overexposed GAPDH has to be shown
3A - 24 hours of cycloheximide has such a dramatic effect on cellular viability and homeostasis that any results are at least doubtful.
3A - at least two independent RNF126 siRNA should be used.
3A - RNF126 and GAPDH have the same molecular weight and in this blot the shape of bands and their width are different - is it the same blot?
Rebuttal Figure R1A - I am missing staining of mTOR - which would correctly answer my question if mTOR downregulation is an indirect result of apoptosis. Additionally to clarify if overexpression of RNF126 does not activate apoptosis and thus downregulate mTOR indirectly - could authors add to results with RNF126 overexpression either cleaved PARP or caspase 3 stainings?
